# Hot Water Treatment as Seed Disinfection Techniques for Organic and Eco-Friendly Environmental Agricultural Crop Cultivation

**Minjeong Kim [1], Changki Shim [1,\*], Jaehyeong Lee [1] and Choeki Wangchuk [2]**

[1] Organic Agricultural Division, National Institute of Agricultural Sciences, Wanju 55365, Korea; kjs0308@korea.kr (M.K.); leeja6311@korea.kr (J.L.)

[2] Department of Agriculture, Ministry of Agriculture and Forests, Gasa 16001, Bhutan; choekiw@moaf.gov.bt

\* Correspondence: ckshim@korea.kr; Tel.: +82-63-238-2554

**Abstract:** Seed is an essential input to sustain agricultural productivity. The expansion of agricultural areas to meet global food demand contributes to the emergence of pathogenic microorganisms that contaminate and infect crop seeds. Conventional technologies in controlling seed-borne diseases are, however, not environmentally sustainable. This inspired the authors to explore existing literature on organic disinfection techniques for crop production. Various integrated seed disinfection techniques for major food crops, including rice, wheat, barley, millet, buckwheat, and sorghum, have been presented in this study. Moreover, the authors explored the potentials of hot water treatment as an alternative treatment method in meeting the ideal seed quality for cultivation.

**Keywords:** cereal crops; seed healthy; physical disinfection method; organic seed

## 1. Introduction

Globally, organic agricultural farmland makes up 1.5% of land (2021), with a record of 72.3 million hectares of organic farmland, including in-conversion areas [1]. For instance, the European Union has established a plan to expand organic agriculture to reduce the use of chemical pesticides and fertilizers and reduce carbon through the Green Deal—under this deal of Farm to Fork and Biodiversity Strategies (European Commission, 2020), the European Commission has set a target of at least 25% of the EU's agricultural land to be under organic farming; also, a significant increase in organic aquaculture by 2030 [2]. In Korea, the production of organic agricultural products has continuously increased since 2015. The eco-friendly agricultural product market is expected to reach KRW 2.45 trillion by 2025 [3].

Food-grade grains are usually bought at a premium price, but making it available in the market requires meeting the quality standards that can be difficult to attain [4]. In cultivating seed-based farm produce, farmers are required to use organic seed unless it is commercially unavailable [5]. Despite the increasing demand for organic seeds for the production of organic foods, organic farmers still utilize conventional (non-organic) seeds [6]. Accordingly, the higher cost of organic seeds influences their unwillingness to buy organic certified seeds [7,8].

Seeds transmit various phytopathogens like bacteria, fungi, and viruses that occur in crops. It is desirable, therefore, to select and cultivate pest-resistant crop varieties including those that can withstand effects of drought, flooding, soil-salinity, and climate change. Seeds determine not only the yield, but also the degree of resistance to diseases, pests, and various environmental disturbances.

Hence, seed treatments provide an effective means of preventing seed-borne diseases [9]. Planting seeds free from seed-borne diseases prevents the introduction of pathogens into the agro-environment and prevents their transfer to soil and water [10].

In the case of the United States Organic Materials Review Institute (US OMRI), the disinfection of organic seeds is defined as follows: "In the case of the US OMRI, organic food production is governed by the National Organic Program (NOP), which can be used by organic farmers to improve the performance of their seeds [11]". The basic principle of organic farming is that the farm uses organically produced seeds. However, they are recommended rather than mandatory.

Therefore, we reviewed the different organic seed disinfection techniques emphasizing the application of hot water disinfection.

## 2. Definition of Organic Seeds

Organic seeds are those harvested from crops grown by applying organic agricultural cultivation methods and materials without the use of synthetic pesticides and chemical fertilizers. Chemically treated seeds may be used when organic seeds are unavailable [12] based on the guideline issued by the International Federation of Organic Agriculture Movements (IFOAM) [13].

The Committee on Food Additives and Contaminants (CODEX) also provided guidelines for the production, processing, labeling, and marketing of organically produced foods: "seeds and vegetative reproductive material should be from plants grown per the provisions of these guidelines for at least one generation or, in the case of perennial crops, two growing seasons [14]".

In Korea, the enforcement regulations of the Enforcement Degree of the Eco-friendly Agriculture and Fisheries Promotion Act stipulate that "organic seeds and seedlings derived from plants grown in accordance with the provisions of the item for at least one generation, or two growing seasons, in the case of perennials".

## 3. Regulations on the Use of Organic Seeds

### 3.1. International Organic Certification Bodies

Organic agriculture requires the use of organic seeds and seedlings of the highest quality possible produced through conventional farming methods without the application of inorganic agricultural inputs like fertilizers and pesticides. The IFOAM and Uganda Organic Certification Company (UgoCert) allow the use of conventional seeds if organic and chemically untreated seeds are not available [15]. In EU and NOP of USA, permits chemically treated seeds and seedlings for phytosanitary purposes if organic seeds are not available. However, KRAV, Naturland, and the Soil Association do not allow the use of chemically treated seeds for organic farming. Worldwide, organic certification bodies have banned GMO seeds and seedlings in organic farming [16]. USDA NOP Organic Seed Regulatory Citation 205.204 seeds and Planting Stock Standard encourages farmers to use 100% organic seeds for their organic production system.

### 3.2. Status and Development Prospects of Organic Seeds

Although Korea has not yet established management regulations for organic seeds, international standards (IFOAM, CODEX) are being applied called *mutatis mutandis*. Now, ordinary seeds can be certified as organic products because of the difficulty to obtain organic seeds within Korea's organic product certification system. Local certification standards are required urgently to match the rapid growth of the domestic organic seed market, albeit small; the self-cultivated organic seeds are exchanged among small-scale indigenous seed clubs [17]. The strict standards for the use of organic seeds are expected to be applied internationally.

Recently, in response to the demand of eco-friendly agricultural farmers for organic seeds: rice, soybean, barley, and wheat have become available as unpollinated seeds according to the National Seed Resources (www.seednet.go.kr, accessed on 9 January 2022) (Table 1).

**Table 1.** Performance of unsterilized seeds supplied by the Korean government.

| Year | Rice Seed (Ton) | | Soybean Seed (Ton) | | Barley Seed (Ton) | | Wheat Seed (Ton) | |
|------|-------|-------------|-------|-------------|-------|-------------|-------|-------------|
| | Total | Unsterilized | Total | Unsterilized | Total | Unsterilized | Total | Unsterilized |
| 2015 | 24,254 | 5238 (21.6%) | 1127 | 192 (17.0%) | 1755 | 293 (16.7%) | 614 | 167 (27.2%) |
| 2016 | 19,470 | 5810 (29.8%) | 1071 | 128 (12.0%) | 1827 | 247 (13.5%) | 509 | 123 (24.2%) |
| 2017 | 21,975 | 6689 (30.4%) | 1082 | 222 (20.5%) | 2542 | 617 (24.3%) | 675 | 314 (46.5%) |

## 4. Importance of Disinfecting Organic Seeds

Unhealthy seeds can reduce the seed quality, serve as plant pathogenic inoculum in the field and cause severe damage to crop production and seed safety and impede international trade [18]. Moreover, the trend in rice consumption worldwide is increasing sharply along with the expansion the world population, specifically in Asia and African countries that generally depend on rice as the main daily calorie source [19]. Hence, disinfecting rice seeds before seeding is an important and indispensable process to prevent infection with seed-borne pathogens, which cause serious damage to yield [20,21].

Table 2 shows the various pathogens infecting plants during cultivation, which may be still dormant along the process. Even if the percentage of infected seeds is low when plant diseases occur in the field, seeds act as the primary source of diseases that can spread rapidly and cause great damage to the crops [22]. Therefore, it necessitates the use of eco-friendly seed disinfection technology to remove pathogens that are attached to the seed surface or exist inside the seeds [23,24].

**Table 2.** Seed-borne diseases and related pathogens of selected food crops.

| Crops | Seed-Borne Diseases | References |
|-------|---------------------|------------|
| Rice (*Oryza sativa* L.) | Blast (*Magnaporthe grisea*) Bakanae disease (*Gibberella fujikuroi*) Brown spot (*Cochliobolus miyabeanus*) | [24] |
| Barley (*Hordeum vulgare* L.) | Covered smut (*Ustilago hordei*) Loose smut (*Ustilago nuda*) | [23] |
| Wheat (*Triticum aestivum* L.) | Stinking smut (*Tilleta caries*) *Helminthosporium* leaf blight (*Helminthosporium sorokiniana, Biopolaris sorokiniana, Cochilobotus sativus*) | [23] |
| Soybean (*Glycine max* (L.) Merr.) | Anthracnose (*Colletotrichum gloeosporioides*) Pod and stem blight (*Diapothe phaseolorum* var. *sojae*) Leaf spot (*Cercospora sojina*) Purple blotch (*Cercospora kikuchii*) Bacterial pustule (*Xanthomonas axnopodis* pv. *glycines*) Viral diseases (AMV, CPMV, SYMMV, SMV, SDV) | [23] |

## 5. OMRI's Organic Seed Treatment Methods

The US OMRI (2010) [11] defines the purpose of seed treatment as to improve the performance of seed. Ensuring the health of organic seeds covers a wide range of treatments, including hot water, biological (microbial) and plant extracts, and bleaching/disinfection. Such treatments can improve seed and seedling health by removing seed-borne pathogens from the seeds.

OMRI (2010) stated that most seed protectants are not an option for organic growers, but organic farmers may use seed treatments such as priming, pelletizing, hot water treatment, and NOP-compliant protective agents to improve seed performance [11].

### 5.1. Priming

Primed seeds are those presoaked with controlled hydration—they germinate faster and more uniformly over a wider temperature range since growth is quiescent; it reduces the likelihood of generating very thick or thin plant stalks. Priming increases the defense mechanisms of germinated seedlings against pathogens—by enhancing the ability of seedlings to absorb enough water necessary in dissolving germination inhibitors in the early stages of germination. Priming is usually performed in conjunction with the pelleting process to protect primed seeds with a shortened lifespan.

Priming is a treatment technology for the purpose of improving germination power, such as shortening the number of days required for germination and uniform germination [25,26]. Seed initiation can increase the antioxidant enzyme content and vigor of seeds.

Wheat seed primed with selenium can also enhance antioxidant levels and the activity of oxidative defensive enzymes 2,3 [27,28]. Rice Seed primed with low selenate concentrations were rice seedling germination and growth were promoted by priming with low selenite concentrations (15–75 mg/kg), but were inhibited by priming with high selenite concentrations (90–105 mg/kg) [29].

Wheat seeds primed with chitosan induced resistance to fusarium head blight disease and improved seed quality [30]. Maize seed priming with two different acidic chitosan solutions improved the vigor of maize seedlings [31]. Rice seed primed with chitosan may accelerate seed germination and improve the tolerance to stress condition of hybrid rice seedlings [32].

### 5.2. Pelleting

Seed pelleting involves the application of a clay coating, which is usually mixed with other clays, to streamline the size, shape, and uniformity of non-round seeds, such as lettuce, carrots, onions, and many herbs and flowers. The pellets make mechanical seeding easy, safe, and accurate by reducing the spacing between sites and the need for labor-intensive sowing. Ideally, the pelletizing material is permeable to oxygen and absorbs water quickly so that when the pellets are submerged in water, they immediately split. Existing pelletizing techniques using synthetic inert materials are not approved for use in organic farming, but there are currently several pelleting materials on the market that are approved for use on organic farms.

Currently eggplant seed coating or pelleting treatment is mainly for vegetables, such as coating for onion seed pest control [33] and improving the germination rate of carrot and tomato seeds [34,35]. It was widely used for seeds.

Seed treatment technology is being applied to improve the seeding efficiency of fine seeds. Rice seeds are known to be attacked by many pathogenic fungi, which is a major cause leading to seed deterioration and degradation of rice grain qualities [36].

The research on physiological and biochemical indexes of grain seed pelleting techniques are extended to wheat [37,38], corn [39–41] and rice [42,43]. $H_2O_2$ seed coating enhanced the germination rate and increased seedling and stem length in the quality protein maize (QPM) variety [41]. Wheat seed coated with chitosan improved chlorophyll content compared to that of the control, and significantly improved the growth index, including germination rate, wet weight, root length, root active, and impacted physiological indices. The field applications of chitosan coating wheat seed showed the increasing yield 13.6% over that of control [38].

## 6. OMRI's Organic Seed Disinfection Technology

Ensuring the health of organic seeds covers a wide range of treatments, including hot water, biological (microbial) and plant extracts, bleaching, and disinfection. Such treatment can improve seed germination performance by removing seed-borne pathogens and preventing the attack of pathogens in the soil.

### 6.1. Hot Water Treatment

The use of hot water treatment to eradicate seed diseases, particularly those caused by phytopathogenic bacteria is well established. This technique works on cabbage, carrots, tomatoes, and peppers, and to some degree, on celery, lettuce, and spinach [44].

The accuracy of temperature and timing is important because the seed's embryo can die in hot water or return to incomplete dormancy in cold water. Therefore, fresh seeds with high vitality should be treated with hot water, as older or low-viability seeds may not respond well to the stress of hot water treatment and may have reduced viability. Moreover, it is recommended to use hot water treated seeds within a season, as it can reduce the vitality of the seeds [44].

### 6.2. Plant Extracts and Essential Oils

The evaluation of plant extracts and essential oils to be used as a seed treatment option is an emerging area of research with few available literatures about their efficacy. For instance, vegetable oils such as thyme, cinnamon, clove, lemongrass, oregano, and garlic oils have the potential to inhibit *Xanthomonas campestris* pv. *campestris*, *Clavibacter michiganensis* subsp. *michiganensis*, *Alternaria dauci,* and *Botrytis aclada*—thyme oil is already being used in Europe to treat seeds [45]. Refined soybean or mineral oil has been shown to reduce molds associated with storage diseases in corn and soybeans [46]. However, additional research is needed to investigate the potential of plant essential oils in inhibiting plant pathogens for the development of seed disinfection techniques.

The garlic tablet, formulated by Integrated Pest Management (IPM) Laboratory, Bangladesh Agricultural University, Mymensingh [47]. Reports on the efficacy of garlic extract and garlic tablet for seed treatment against seed-borne fungi and to increase yield of different crops are available in Bangladesh [4–50].

Treatment of rice seeds with garlic was found to be effective to decrease seedling diseases viz. brown spot, blast, bakanae, root rot and seedling blight [49].

All concentrations (1:3, 1:4, 1:5 and 1:6 ratio diluted in distilled water) of the garlic tablet significantly decreased the prevalence of seed-borne fungi of cucumber, occurrence of abnormal seedlings and rotten seeds as compared to untreated control [47].

Garlic extract showed fungicidal activity on the endogenous fungal contamination of the barley [51], wheat seeds [52] and particularly reduced the degree of disease caused by *Bipolaris sorokiniana* and *Drechslera tritici*-repentis. Allicin in garlic juice corrected the poor germination of wheat seeds caused by natural microorganisms of grain. Growth promoting activities of garlic extract on wheat seedlings was reported. Interestingly, the inoculum on naturally infected wheat seeds could be reduced with garlic juice as a seed dressing biofungicide, before sowing [52].

In this aspect seed treatment with botanicals may be a safe option in controlling seed borne pathogens. Uses of plant extracts in controlling pathogens are now found to be promising and successful against certain fungal pathogens [53–57].

Hossain and Schlosser (1993) [58] reported sound fungicidal effect of Neem (*Azadirachta indica*) extracts against *Bipolaris sorokiniana*.

The attractiveness of plant extracts, garlic tablets [47–55], and neem leaf extract [50,54–58] is that they significantly reduced the total seed-borne fungal infections in the population of individual six target pathogenic fungi: *Agrostis tenuis*, *Bipolaris sorghicola*, *Botrytis cinerea*, *Crinum graminicola*, *Curvularia lunata*, and *Fusarium moniliforme*. Specifically, garlic tablets and neem leaf extract reduced over 90.0% seed-borne infection of *B. sorghicola* and *C. lunata*. Additionally, germination percentage increased above 80.0% in all the treatments compared with conventional treatment in sorghum [50].

### 6.3. Bleach Disinfection

Bleach (sodium hypochlorite) may be used to disinfect the seeds instead of hot water. Bleach removes pathogens from the seed surface, but not pathogens underneath the seed husk. Sodium hypochlorite is permitted for disinfection in water on organic farms if the

maximum residual contamination levels of safe drinking water practices are not exceeded; the currently acceptable chlorine content is 4 ppm [59].

The sodium hypochlorite treatment, which is more compatible than soaks with commercial seed sterilizants, inexpensive, easily available, simple procedure and has less potential for injury to seed, should be highly useful [60,61].

Barley seeds were treated with 500 mg/L chlorine dioxide, and $ClO_2$ significantly increased the fresh weight and total length of barley roots, higher than that of control [62].

Application of 2 to 4% household bleach for 24 h to rice seed was recommended for increased germination percentage, mean germination time and germination rate, root and shoot length, and root and shoot dry weight [63].

Modification of sodium hypochlorite treatment combined with warm water might be an important tool for solving phytosanitation of teliospores of *Tilletia controversa* on wheat grain and seed [64].

## 7. Eco-friendly Seed Disinfection Technology for Food Crop Seeds

### 7.1. Disinfection of Rice Using Hot Water

Eco-friendly seed disinfection technologies using steam, microwaves, ultraviolet light, or low-temperature plasma, along with cold water immersion, have been developed, but in general, cold water immersion technology is more widely used. Target crops include rice, wheat, oats, barley, and sorghum. In Table 3, the treatment temperature is slightly different for each crop, but disinfection is generally carried out at 51–55 °C for 7–20 min. As a method of mass disinfection for quantities of seeds exceeding 1 ton, disinfection technology using steam at high temperature (60–65 °C).

**Table 3.** Eco-friendly seed disinfection technology for rice.

| Target Disease | Applied Technique | References |
|---|---|---|
| Blast (*Magnaporthe grisea*) | Soak in cold water (18–20 °C) for six hours, then in 50 °C hot water for two minutes | [65] |
| Leaf spot (*Helminthoporium oryzae*) | Soak in 51 °C hot water for seven minutes | |
| Bacterial Grain Rot (*Burkholderia glumae*) | Soak in cold water for four hours then soak in 65 °C hot water for 5 min | [66] |
| Bakanae (*Gibberella fujikuroi*) | Pretreatment in 50 °C hot water for 2–3 min then soak in 58 °C hot water for 15 min | [67] |
| | Soak in 60 °C hot water for 10 min | [68] |
| | Soak in 1% sodium hypochlorite | [69] |
| | Soak in 60 °C hot water for 10 min then in 0.02% lime sulfur mixture for 24 h | [70] |
| | Soak in 60 °C hot water for 10 min, then in 1% loess sulfur for 48 h | [68] |
| | Soak in salt solution (specific gravity, 1.13 g/L, 5.0 kg/20 L) for 5 min | [71] |
| | Soak in 1% garlic extract for 24 h | [72] |
| Bakanae (*Gibberella fujikuroi*)/ Bacterial grain Rot (*Burkholderia glumae*) | Soak in 60 °C hot water for 10 min then or in 0.2% copper hydroxide (Cu(OH)$_2$) for one hour | [70] |
| Bakanae (*Gibberella fujikuroi*) | Soak in 30 °C hot water for 48 h | [71] |

One technique to reduce rice blast disease caused by *Magnaporthe grisea* is via immersion in cold water for 6 h, then immersion in 50 °C hot water for 2 min. In addition, rice

seeds contaminated with *Helminthosporium oryzae* are immersed in 51 °C hot water for 7 min [65].

Among the various diseases that occur in rice, bakanae disease is a representative seed-communicable disease, and its incidence is increasing. Therefore, control measures are required, and there is a need to develop an alternative control technology to chemical pesticides. When seedlings with rice bakanae disease are transplanted, the growth is affected, and the yield and quality of the rice are reduced. It is known that infected seeds become a continuous source of transmission of the disease.

The occurrence of bakanae disease pathogens with resistance to chemical pesticides that have been used in conventional cultivation has been reported. Even in organic cultivation, bakanae disease is an important disease that is a problem from sowing to harvest. Various rice seed disinfection technologies have been developed and are likely to be used on farms in the future [66–71].

Recently, the occurrence of seed-borne diseases, such as fungal disease and bacterial blight, have increased due to abnormally high temperatures and the expansion of eco-friendly cultivation areas. Therefore, the seed disinfection technology such as hot water immersion and organic agricultural materials—sulfur, copper, vegetable oil, plant extracts, or Bordeaux liquid are developed and effectively used in Korea (Table 3).

The typical seed disinfection technique used by farms is hot water immersion, which lowers the occurrence of pathogens and improves the germination rate. In addition, disinfection using salt, lactic acid, and organic materials (lime sulfur, Bordeaux liquid, plant extracts) is currently being developed and utilized.

Rice seeds are immersed in 60 °C water for 10 min. Seeds are then sterilized by immersion in lime sulfur compound diluted solution for 24 h.

So et al. (2017) [68] reported the rice seed disinfection efficacy of loess-sulfur for the suppression of bakanae disease caused by *Fusarium fujikuroi*. Rice seeds were treated at different concentrations and combinations of loess-sulfur, soaking time, and temperature.

To prevent rice bacterial blight, the rice seed is immersed in cold water for 4 h, then immersed in hot water at 65 °C for 5 min or in hot water at 60 °C for 10 min, and then immersed in 0.2% copper hydroxide solution for 1 h [70].

### 7.1.1. Salt Gravity Method for Buckwheat and Rice seed

When whole rice seeds are selected for the saltwater method, the seed disinfection effect increases by 20–30% by dissolving 5 kg of salt in 20 L of water for buckwheat and rice. To prevent tall leg disease, seeds are immersed in brine with a specific gravity of 1.13 in 20 L of water and 5 kg of solar salt for 5 min to select only the seeds that settled to the bottom (Table 4) [73].

**Table 4.** Restriction and growth of rice seedlings after being treated with salt with different specific gravity [73].

| Variety | Parameter | Salt Specific Gravity | |
| --- | --- | --- | --- |
| | | **More than 1.06 g/L** | **More than 1.13 g/L** |
| Nampeng rice | Restriction of growth (%) | 94.5 | 56.1 |
| | Plant height (cm) | 26.2 | 26.1 |
| | Leaf number (No.) | 4.1 | 4.1 |
| | Dry weight (mg/plant) | 30 | 31 |
| Daeripbey 1 ho | Restriction of growth (%) | 95.4 | 78.6 |
| | Plant height (cm) | 31.8 | 32 |
| | Leaf number | 4.1 | 4.1 |
| | Dry weight (mg/plant) | 42 | 43 |

20-day seedlings grown at an average temperature of 24 °C, 20-day seedlings grown.

When rice seeds are sown, it is important to use a salt specific gravity to secure an appropriate rice seedling rate and use an appropriate amount of salt to match this salt

gravity. In the case of rice seed by dissolving with 5 kg of salt, there is no sinking seed, so it is necessary to change the amount of salt to 4.24 kg/20 L (Table 4).

Saltwater shielding increases the rate of adult seedlings and dry seedlings. In cold areas where various disturbances easily occur, germination and early growth should be promoted to achieve the ideal effect. If rice seeds are soaked in salt water for a long time, germination can be easily impaired, thus the need to wash it with fresh water immediately. Saltwater can also greatly reduce the occurrence of bakanae disease [71].

### 7.1.2. Disinfection of Rice Seeds Using Hot Water Treatment

As an eco-friendly seed disinfection method, the hot water disinfection method is being developed and applied by rice farmers. However, some varieties of rice have lower germination rates depending on the immersion conditions. Among general rice cultivars, eight varieties (Gounbyeo, Dongjin 1 ho, Seoan 1 ho, Pungmibyeon, Samgwangbyeo, Shinunbong 1 ho, Ungwangbyeo, and Ilmibyeo) have germination rates that significantly decrease when immersed at 60 °C for more than 10 min (Table 5).

**Table 5.** Conditions for seed sterilization by hot water treatment according to rice variety [74].

| Treatment | Rice Varieties |
|---|---|
| 60 °C, 10 min | Seoan 1 ho |
| 60 °C, 15 min | Dongjin 1 ho, Seoan 1 ho, Pungmibyeo |
| 60 °C, 10 min | Gounbyeo, Dongjin 1 ho, Samgwangbyeo, Seoan 1 ho, Shinunbong 1 ho, Unkwangbyeo, Ilmibyeo, Pungmibyeo |
| 62 °C, 10 min | Baegjinju 1 ho, Suranyeo, Shindongjin, Younganbyeo, Taeseongbyeo, Hanambyeo, Haepyeongbyeo, Gounbyeo, Seoan 1 ho, Shinunbong 1 ho, Unkwangbyeo |
| 62 °C, 15 min | Gopumbyeo, Saegyehwabyeo, Saechuchengbyeo, Surabyeo, Shindongjin, Younganbyeo, Odaebyeo, Junamjosaeng, Chengdam, Taeseongbyeo, Pungmi 1 ho, Haepyeongchal, Hopumbyeo, Gounbyeo, Dongjin 1 ho, Samgwangbyeo, Seoan 1 ho, Shinunbong 1 ho, Unkwangbyeo, Ilmibyeo, Pungmibyeo |

The treatment method involves sterilizing the dry seeds by immersing them in hot water at 60 °C for 10 min, then immediately removing them and cooling them in cold water for 30 min. The organic materials will be processed for 24 h at 30 °C, and the rice seeds will be sown. This treatment has also been shown to have an inhibitory effect on rice blight [74].

The varieties followed a similar trend in response to the hot water treatments. The temperature tolerance limit of an Indica and a Japonica variety was observed to be 55 °C hot water temperature regardless of exposure period [75].

### 7.1.3. Disinfection of Colored Rice Varieties Using Hot Water

Colored rice is sensitive to the temperature and time of immersion in hot water. However, it has been reported that varieties of colored rice tend to require a longer soaking period for germination than other varieties of rice. In the case of seed disinfection using germination at a farmhouse sown under conditions similar to those of ordinary rice, after 48 h at 30 °C low germination rate of seedlings was observed depending on the variety. Among the colored rice varieties, healthy red rice and red pearl can be sown immediately after sterilization at high temperature disinfection. A safe sowing period is required after seed disinfection (Table 6).

**Table 6.** The safe incubation period of water temperature after disinfection of colored rice varieties is based on a germination rate of 80% [74].

| Safety Soaking Time (Hours) | Water Temperature (°C) | | |
|---|---|---|---|
| | **15** | **18** | **21** |
| 0~1 | Jukjinju, Geonganghongmi, Heugkwang | Jukjinju, Geonganghongmi, Heugkwang | Jukjinju, Geonganghongmi, Heugkwang |
| 2~3 | - | Heugnam, Heugseol, Shintoheugmi, Shinmyeongheugchal, Boseogheugchal | Heugnam, Heugjinju, Heugseol, Shintoheugmi, Shinmyeongheugchal, Boseogheugchal |
| 4~5 | Shinmyeongheugchal, Boseogheugchal, Shintoheugmi, Josaengheugchal | Heughyang, Shinnongheugchal, Josaengheugchal | Heughyang, Shinnongheugchal, Josaengheugchal |
| 6~7 | Heugnam | Heugjinju | - |
| 8~9 | Heughyang, Heugseol, Shinnongheugchal, Heugjinju | - | - |

### 7.1.4. Disinfection of Rice Seeds Infested with Bacterial Blight

Bacterial rice blight is a seed-borne disease that occurs in rice grains and leaf sheaths. Ears that are infected have a pale red color and stand without bowing their heads. In bacterial blight-infected rice plants, the growth of the embryo is stopped, and it becomes chaff. The seed disinfection effect was tested after immersing the seeds in cold water for 4 h, then immersing them in hot water at a temperature of 50–65 °C for 5 min to sterilize the bacterial blight-infected rice seed. When sowing was carried out after treatment, the germination rate was maintained at 78.3% or more, and it was shown that the occurrence of bacterial blight could be reduced [70,76].

Hot water disinfection is an easy and effective seed disinfection technique that does not require the use of chemicals [75]. The typical condition of hot water treatment is not always sufficient to disinfect certain pathogens. The higher temperatures repressed the germination rates of several rice cultivars, especially indica rice types [76].

The seeds of japonica cultivars show greater heat tolerance compared to indica cultivars. For example, the germination rate of the japonica cultivar 'Hitomebore' was more than 90% when the seeds were treated with hot water at 67.5 °C for 10 min [77]. Kashiwagi et al. (2017) [78] reported that the average germination rate of japonica rice cultivars was 52.2%, which is higher than that of indica rice cultivars (31.5%) under hot water treatment at 69 °C for 10 min.

### 7.1.5. Disinfection of Rice Seed Using Organic Agricultural Materials and Hot Water Treatment

The varying effects of disinfection treatments in organic rice seed calls for the improvement of seed disinfection technologies. The seed disinfection effect was investigated when hot water immersion technology and organic agricultural materials such as sulfur, copper, vegetable oil, plant extract, and Bordeaux liquid were added (Table 7).

**Table 7.** Rice seed disinfection effect by hot water treatment and combined treatment with organic agricultural material [79].

| Treatment | Treatment Content (L) | Inhibition of Bacterial Grain Rot | Incidence of Bakanae Disease (%) | Seedling Emergence (%) | Control Value (%) |
|---|---|---|---|---|---|
| Hot water treatment (60 °C, 10 min) | - | - | $0.44 \pm 0.05$ b * | 88.7 | 95.7 |
| Hot water treatment combined with a lime sulfur mixture | 20 mL | - | $0.10 \pm 0.02$ a | 84.7 | 99 |
| Hot water treatment combined with copper hydroxide (Cu(OH)$_2$) | 2 g | + | $0.10 \pm 0.02$ a | 85.4 | 99 |
| Hot water treatment combined with cinnamon and castor oil | 1 mL | - | $0.10 \pm 0.01$ a | 84.5 | 99 |
| Hot water treatment combined with castor and tea tree extract | 5 mL | - | $0.10 \pm 0.02$ a | 84.7 | 99 |
| Hot water treatment combined with Bordeaux mixture | 20 g | + | $0.15 \pm 0.05$ a | 87.7 | 98.6 |
| Control | - | - | $10.36 \pm 0.25$ c | 83.4 | - |

* Means followed by the same letter(s) in a column did not differ significantly at 1% level by DMRT.

### 7.2. Disinfection of Cereal Crops Using Hot Water

Organic cereal crops occupy about one-third of the total organically cropped area in the EU, ranging from 18% in the Netherlands to 63% in Lithuania [4]. One of the main problems in organic farming is the seed health of cereals [7,8]. Therefore, it is necessary to develop pesticide-free seed disinfection technologies to promote stable germination and improve seeding rate of selected overseas cereal crops by sterilizing seed-infecting fungi for eco-friendly production (Table 8).

The occurrence of sorghum ear mold disease among grains has increased due to abnormal weather conditions. When seeds infected with sorghum ear mold disease are sown, the germination and piling rates drop sharply. Therefore, it is necessary to develop a pesticide-free seed disinfection technology to improve the stable germination and seeding rate of sorghum by eliminating seed-infecting fungi for eco-friendly agricultural production.

**Table 8.** Eco-friendly seed disinfection technology for overseas cereal crops.

| Crop | Target Disease | Applied Technique | References |
|---|---|---|---|
| Sorghum (*Sorghum bicolor* L.) | Grain mold (*Fusarium moniliforme*) | Soak in 10% wood vinegar for two hours | [79] |
| | | Soak in 60 °C hot water for 10 min | [79] |
| Foxtail millet (*Setaria italica* Beauvois) | Bakanae (*Gibberella fujikuroi*) | Soak in salt solution (specific gravity, 1.040 g/L, 4.5 kg/20 L) for 5 min | [80] |
| Millet (*Panicum italicum* L.) | Grain mold (*Fusarium moniliforme*) | Soak in 10% wood vinegar for two hours | [81] |
| | Downy mildew (*Sclerospora graminicola*) | Soak in cold water (18–20 °C) for six hours, then in 50 °C hot water for two minutes | [82] |

**Table 8.** *Cont.*

| Crop | Target Disease | Applied Technique | References |
|---|---|---|---|
| Wheat (*Triticum* aestivum L.) | Loose smut (*Ustilago muda*) | Soak in warm water (21 °C) for five hours then soak in 49 °C hot water for one minute and in 52 °C hot water for 11 min or in 55 °C hot water for 10 min | [83] |
| | | Soak in 41 °C Hot water for 300–360 min or in 49 °C hot water for 90–120 min | [83] |
| | Fusarium head blight (*Fusarium graminearum*) | Exposure to 2.45 GHz microwave for 20 min | [84] |
| | Fusarium leaf spot (*Fusarium nivale* f. sp. *graminicola*) | Exposure to water vapor at 63 °C for 5 min | [85,86] |
| Oat (*Avena sativa* L.) | Septoria avenae blotch (*Parastagonospora avenaria* f.sp. *avenaria*) | Exposure to water vapor at 59 °C for 5 min | [85,86] |
| Barley (*Hordeum vulgare* L.) | Loose smut (*Ustilago muda*) | Exposure to water vapor at 65 °C for 5 min | [85,86] |
| | | Hot water treatment (41 °C, 300 min or 49 °C, 90 min) | [86] |
| | Fusarium head blight (*Fusarium graminearum*) | Dipping in Cedomon solution at a rate of 7.5 mL/kg of *Pseudomonas chlororaphis*, strain MA 342 | [87] |

To improve germination and hair growth rate that is reduced due to ear fungus on sorghum, sorghum seeds were disinfected at intervals of 1, 2, and 4 h with a dilution of wood vinegar solution 2, 5, and 10 times. After disinfecting the seeds for 2 h, it was found that the seeding rate was increased by 10.8% [81]. The quality of wood vinegar was certified by a national institution (Act on the Promotion of Forestry and Mountain Villages, 2011). At 2.5 tar is 1.5% or less, and the refractive index is 3.5 or more [81].

Millet downy mildew disease can be reduced by immersion in hot water at 50 °C for 2 min [77].

Winter et al. (1994) [83] reported that the hot-water treatments for the control of loose smut caused by *Ustilago muda* in wheat and barley give a stable result over a range of treatment conditions, time, and temperature combinations. The soaking period recommended for wheat seed ranges from 90 or 120 min, at 49 °C (120 °F) to 300–360 min at 41 °C (105 °F); for barley seed ranges from 90 or 300 min at 49 °C (120 °F) to 41 °C (105 °F).

Wheat fusarium head blight (FHB), smut, leaf stripe, and net blotch disease are transmitted via seeds in barley. Strong microwave (2.45–5.7 GHz) electric field treatment against wheat seed-borne pathogens, such as the causal agents of seedling and adult plant foot and root rot and FHB, has produced conflicting and inconsistent efficacy [84].

Wheat brown leaf disease is eliminated by exposure to steam at 63 °C for 5 min, while oat leaf disease can be eliminated by exposure to steam at 59 °C for 5 min, and barley rot can be eliminated by steam exposure at 65 °C for 5 min [85,86].

Forsberg et al. (2005) [86] reported that the optimum strategies for treatment time and air humidity were quick heating (59–65 °C) with humid air for a short time (almost 5 min), immediately followed by rapid cooling. This procedure involves the selective heating of the external layers of the seeds of wheat, oat, and barley, and targets important cereal seed-borne pathogens.

The main effect of these methods is seed sanitation, while not affecting the growing part of plants after seed germination. Liatukas et al. (2019) [87] reported that bacterial treatment with Cedomon increased the yield of two varieties of barley, Luokė and Alisa DS, by an average of 3%. This treatment improved neither seed health nor crop stand density in the Luokė variety; however, it had a positive effect on the seeds of Alisa DS in the experimental years 2014 and 2015.

To suppress the occurrence of fungal and bacterial pathogens and ensure stable germination, sterilization by immersion in hot water at 60 °C for 10 min under hot water immersion conditions showed a germination rate of 86.4% in sorghum (Table 9).

**Table 9.** Percentage of bacteria detection and germination of sorghum under hot water immersion treatment conditions [88].

| Treatment Condition | Detection Percentage of Microbes (%) | Percentage of Germination (%) |
|---|---|---|
| 60 °C, 10 min | 7.8 b | 86.4 a * |
| 60 °C, 15 min | 6.7 b | 73.3 b |
| 60 °C, 20 min | 4.3 c | 55.7 c |
| Control | 14.4 a | 88.5 a |

* Means followed by the same letter(s) in a column did not differ significantly at 1% level by DMRT.

## 8. Conclusions

Approximately 40% of economic losses occur every year due to diseases, weeds, and grain infection of major crops, resulting in great economic damages, increase in chemical pesticide production, and pollution of the agro-environment including water and agricultural products [89].

The planting materials such as seeds and seedlings should be conventionally grown or untreated with a chemical to be considered organic. EU and US permit chemically treated seeds and seedlings for phytosanitary purposes if organic seeds are unavailable. Korea has not yet established management regulations for organic seeds, but presently practices the international standard of IFOAM and CODEX.

However, looking at the case of organic certification revocation in Korea, it was found that there is a direct disadvantage in farms that organic certification is unintentionally revoked using conventional seeds treated with pesticides.

Disinfection technology for the seeds and seedlings is vital to remove pathogens. Most seed protectants are not an option among organic growers. However, there are existing seed treatment practices such as priming, pelletizing, and the use of hot water or NOP-compliant protectants, which can be adopted by organic farmers to improve seed germination performance and protect against harmful pathogens. Plant extracts and organic materials can be also used as seed protectants.

Hot water disinfection is emerging as one the best alternative to treat phytopathogenic bacteria, viruses, and fungi, which works well with rice, buckwheat, wheat foxtail millet, millet, oat, barley, and sorghum. The temperature and time combination of hot water treatment is standardized for different crops, which helps to protect the seed's embryo—it can be a breakthrough and make organic seeds affordable. However, seeds of high vitality are still required to achieve optimal germination. Hot water disinfection technology, as a clean, simple, and low-cost technique, can be disseminated to developing countries with high incidences of pesticide utilization. Rice seed disinfection with loess-sulfur, which is popular to farmers in the Republic of Korea, can be promoted to farmers who either practice hot water-disinfector or seed-disinfector. These alternative techniques are recommended to enhance the quality of seeds and promote eco-friendly seed disinfection technologies.

Recently, researchers are giving more emphasis on sustainable technology for environmentally friendly agriculture. So, it has become necessary to find out ecologically sound, economically viable, culturally appropriate and socially adoptable technology.

**Author Contributions:** Conceptualization, M.K. and C.S.; methodology, M.K.; software, J.L.; formal analysis, M.K.; investigation, M.K.; resources, M.K.; data curation, C.W.; writing—original draft preparation, M.K. and C.S.; writing—review and editing, C.S. and C.W.; visualization, M.K.; supervision, C.S.; project administration, C.S.; funding acquisition, C.S. All authors have read and agreed to the published version of the manuscript.

**Funding:** This research received no external funding.

**Institutional Review Board Statement:** Not applicable.

**Informed Consent Statement:** Not applicable.

**Data Availability Statement:** Not applicable.

**Acknowledgments:** This study was carried out with the support of the Research Program for Agricultural Science & Technology Development (Project No. PJ01475402) funded by the Rural Development Administration in the Republic of Korea in 2022.

**Conflicts of Interest:** The authors declare no conflict of interest.

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
