# Peer review of "Hot Water Treatment as Seed Disinfection Techniques for Organic and Eco-Friendly Environmental Agricultural Crop Cultivation"

_agriculture, doi:10.3390/agriculture12081081_

Round 1

Reviewer 1 Report

     Seed is a basic and important input for susutainable agricultural  productivity and safe production of the food crops. Conventional technologies for seed-borne disease magement have limitations in efficiency and environmental safety. This manscript reviewed various methods of integrated seed disinfection techniques for major food crops, it wil be good for the establishment of organic agriculture. Minor reviews are needed as the following:

   1. Title: Organic Seed Disinfection Techniques  for SafetySafe Production of Eco-friendly Environmental Agricultural CropsCrop Cultivation.

     2. Check and revise some words in this manuscript, such as Chemicals pesticides , should be Chemical pesticides

     3. It seems that Line 502-506  is the same with Line507-513.

Author Response

And the opinions of the judges were reflected as follows:
1. The title of the thesis has been revised focusing on the hot water immersion method.
2. We have checked that all references are relevant to the contents of the manuscript.
3. We have a try to revision the manuscript using the “Track Changes” 
4. We have provided a rebuttal letter to explain, point by point, the details of the revisions to the manuscript commented by referees’ comments. 

Reviewer 2 Report

1)     the authors use only narrow area of cereal seeds disinfection. Some words on seeds disinfection technologies of other agricultural crops are desirable. Indicate similarities and differences between the cases

2)     According to literature data there are 3 main methods of ecologically friendly disinfection: washing, hot water treatment and non-thermal gas-plasma utilization. The authors speak only about the second one. As the work is a review it is necessary to indicate the place of hot water treatment in this list of methods

3)     Furthermore, it seems highly desirable to expand the review and include appropriate data on microalgae application. Please add more information about garlic tablets, the information about WO patent ‘seeds disinfection method’ patent WO2016\198644 (15.12.2016)

4)     It is desirable to compare different methods of disinfection, their efficiency, drawbacks and benefits, objects of treatment

5)     Too much volume of the text is devoted to definitions of organic seeds, disinfection, etc and decrees of various departments- really it is rather tedious to read such an information

6)     It is desirable to explain enormous variations in seeds treatment technologies (time, temperature, etc) but not just describe various examples, conclusion is necessary concerning the difficulties of disinfection process optimization

7)     By the way, does ethanol treated seeds  refer to organic seeds? (see Yu Si,Yakupjan Haxim, Lei WangOptimum Sterilization Method for In Vitro Cultivation of Dimorphic Seeds of the Succulent Halophyte Suaeda aralocaspica Horticulturae 2022, 8(4), 289)

8)     There are repetitions in the text:

lines 652-656: ‘Ensuring the health of organic seeds covers a wide range of treatments, including hot water, biological (microbial) and  plant extracts, and bleaching/disinfection. Such treatments can improve seed and seedling health by removing seed-borne pathogens from the seeds.’

Lines 680-683: ‘Ensuring the health of organic seeds covers a wide range of treatments including hot water, biological (microbial) and plant extracts, bleaching, and disinfection. Such treatment can improve seed germination performance by removing seed-borne pathogens and preventing attack of pathogens in the soil.

9)     Line 737 ‘Out of the six target pathogenic fungi, garlic tablets and neem leaf extract gave over 90.0% reduction in seed-borne infection of B. sorghicola and C. lunata. These increased germination percentages (> 80.0%) with all the treatments compared with conventional treatment [28].’- add citation for garlic tablets and neem leaf extract. I am not sure that [28] is a correct one,

10) As the review is devoted to cereal seeds may be it is better to delete the information about potatoes??

Recommendations: from a practical point of view it is extremely interesting for the reader to get the whole picture of the problem and especially of disinfection methods used. That is why I would recommend to expand the review and make a comparison of different approaches for seeds disinfection. That will make the work more valuable and interesting for the reader.

Author Response

(The authors gave the same response as above.)

Reviewer 3 Report

- The concept of seed desinfection is very broad, as part of your review, only a number of techniques are mentioned, but not as a girth of all.

- The abstract is very succinct. The beginning of the abstract contains general phrases that do not reveal the essence of the review. The text of the abstract contrasts the treatment of seeds with hot water and all other methods. Highlighting the treatment of seeds with hot water as a dominant will contradict the title of the article.

- Few keywords. Of course, this is not a remark, but I would advise the authors to expand the list of keywords, as well as pick up those keywords that will not be found in the title and abstract. This approach would allow to expand the boundaries of the issuance when requested in search engines.

- There is no logical transition in the introduction from describing the growth rates of organic agriculture to seed sterilization techniques. In the introduction, the authors do not focus on specific sets of seed disinfection techniques used by Eco-friendly Environmental Agriculture. Attention should be paid to a brief assessment of their specifics, as well as experience in using.

- Despite the fact that the article is a review, the introduction would greatly embellish the existence of a hypothesis.

- A number of abbreviations are deciphered several times in the course of the manuscript. Check, please.

For example United States Organic Materials Review Institute (US OMRI). At the same time, some names occur once, I can assume that there is no need to introduce an abbreviation. In general, it should be noted that the introduction of a separate section for the abbreviation would greatly simplify the navigation through the text.

- Line 295-302 Large piece of text, not accompanied by a reference.

- Table 1. Performance of unsterilized seeds supplied by the Korean government

A number of data are converted as totals, and other data (in particular %) as averages. There is some inconsistency in the designation.

- 6. OMRI's Organic Seed Disinfection Technology

I would recommend the authors to expand this section. It seems to me the key in the context of this review. Emphasize the advantages and disadvantages of each processing method. Application specifics. Application restrictions. It is possible to give some examples of real use in the framework of research work and / or agricultural reports.

- Some confusion with subparagraphs of section 7

7. Eco-friendly Seed Disinfection Technology for Food Crop Seeds

7.1. Disinfection of Potato and Rice using Hot Water Treatment

7.2.1.1. Salt Gravity Method for Buckwheat and Rice seeds

7.2.1.2. Disinfection of Rice Seeds Using Hot Water Treatment

7.2.1.3. Disinfection of Colored Rice Varieties using Hot Water Treatment

7.2.1.4. Disinfection of Rice Seeds Infested with Bacterial Blight

7.2.1.5. Disinfection of Rice Seed using Organic Agricultural Materials and Hot Water

treatment

7.1.2. Disinfection of Cereal Crops using Hot Water Treatment

There is no logical connection between the points, or I cannot catch it. In my opinion, it is necessary to group points by plants or group them according to the nature / specifics of processing.

- In a number of tables, for example table 9 and table 11, there are letters opposite the values. I assume that they mean the reliability of the differences between the variants of the experiment. However, the data cannot be presented in this way; an error scale must be introduced. Otherwise, it turns out that, for example, there are no differences between 0.10 and 0.15 (Table 9), and from a mathematical point of view, this is absurd.

- In conclusion, the article considers the extremely important aspects of the development of eco-friendly agriculture. However, the manuscript is not devoid of obvious shortcomings, it is poorly structured and the authors do not always provide sufficient arguments for the choice of specific processing tools. I am sure that the article can be improved.

Author Response

(The authors gave the same response as above.)

Round 2

Reviewer 2 Report

Minor comments

Line 51- insert ‘products’

Line 64 ‘Accordingly, the higher cost of conventional seeds influences their unwillingless to buy organic certified’-is it right: ‘conventional seeds?

Line 78 change ‘prevent’ to ‘prevents’

line 154 style: delete repetition ‘Some certification bodies permit the use of conventionally grown, untreated seeds and seedlings when organic seeds or seedlings are unavailable. The IFOAM and Uganda Organic Certification Company (UgoCert) allow the use of conventional seeds if organic and chemically untreated seeds are not available [15]. In EU and National Organic Program (NOP) of the United States of America (USA), permits chemically treated seeds and seedlings for phytosanitary purposes if organic seeds are not available.

Table 3- delete the first column and change the title of the Table from”… for overseas food crops’ for ‘rice’

Line 455 ‘germination rates that significantly decreases when immersed…’ change to ‘germination rates that significantly decrease when immersed..’

Line 607- add ‘in sorghum’

Line 639- delete ‘potato’

Author Response

Reviewer 2: Minor comments
line 51- insert ‘products’
Insert “products”
line 64 ‘Accordingly, the higher cost of conventional seeds influences their unwillingless to buy organic certified’-is it right: ‘conventional seeds?
  organically produced

line 78 change ‘prevent’ to ‘prevents’
 change as “prevents”
line 154 style: delete repetition ‘Some certification bodies permit the use of conventionally grown, untreated seeds and seedlings when organic seeds or seedlings are unavailable. The IFOAM and Uganda Organic Certification Company (UgoCert) allow the use of conventional seeds if organic and chemically untreated seeds are not available [15]. In EU and National Organic Program (NOP) of the United States of America (USA), permits chemically treated seeds and seedlings for phytosanitary purposes if organic seeds are not available.
 deleted the paragraph “Some certification bodies permit the use of conventionally grown, untreated seeds and seedlings when organic seeds or seedlings are unavailable.”
Table 3- delete the first column and change the title of the Table from”… for overseas food crops’ for ‘rice’
 Deleted first column and changed the title as “Table 3. Eco-friendly seed disinfection technology for rice.

Line 455 ‘germination rates that significantly decreases when immersed…’ change to ‘germination rates that significantly decrease when immersed..’
 change “decreases” to “decrease”
Line 607- add ‘in sorghum’
 insert “in sorghum” at the end of that sentence
Line 639- delete ‘potato’
 delete “potato” in crop

Reviewer 3 Report

The authors have done a great job to complete the manuscript. I am satisfied with the changes made and believe that the article can be published in this special issue.

Author Response

Thank you.

First of all, we had this MS fully re-corrected for spelling, sentences and grammar of the MS as a whole through PICCARD's original English text correction system

This manuscript is a resubmission of an earlier submission. The following is a list of the peer review reports and author responses from that submission.

Round 1

Reviewer 1 Report

I read with interest the manuscript entitled „„Organic Seed Disinfection Techniques for Safety Production of  Eco-friendly Environmental Agricultural Crops Cultivation”.

It is an interesting summary of researches on the use of hot water in disinfection of rice, wheat, barley, soybean and potato seeds.

Unfortunately, the title of the manuscript does not match its content. It should be modified to include the use of hot water in disinfection, as it is the most important part of the manuscript.

The manuscript should state the purpose of the paper at the end of the introduction.

A very large part of the conclusions is suitable for the introduction.

More attention should be given to the use of hot water in the conclusions.

Author Response

All authors of this MS accept opinions pointed out by three reviewers.
First of all, we had this MS fully corrected for spelling, sentences and grammar of the MS as a whole through MDPI (No. 41770)'s original English text correction system.
And the opinions of the judges were reflected as follows:
1. The title of the thesis has been revised focusing on the hot water immersion method.
2. In conclusion, the importance of hot water treatment and the need for additional research on other crops were mentioned in the future.
3. Scientific names are indicated in accordance with the International Nomenclature of Plants.
4. The scientific name of Helminthosporium oryzae was modified.

Reviewer 2 Report

This is an important effort in terms of setting the present knowledge and understanding of how to deal with seed disinfection technologies. This may or may be not appliable to every country and crop. However, you effort and the description and results of the available techniques is rather interesting. 

Author Response

(The authors gave the same response as above.)

Reviewer 3 Report

The manuscript need some revisions:

  • I suggest that the scientific names of cultures be standardized, including the author as established by the International Code of Botanical Nomenclature in all Tables, e.g. Oryza sativa L (Table 2);
  • Page 281 – Whenever the scientific name appears for the first time in the text, it must be written in full, including the author. Afterwards, you can abbreviate it. Please review the entire text.
  • In Table 3, Helminsoporium oryzae is mentioned. Wouldn’t the correct one be Helminthosporium oryzae?
  • It recommended that authors make a careful reading in order to correct problems, such as those mentioned above.

Author Response

(The authors gave the same response as above.)

Round 2

Reviewer 1 Report

After reviewing the final version of the maunuscript, I see that the authors have put a lot of effort into improving the manuscript.  In my opinion, this manuscript is a very interesting study and I recommend publishing it in the Agriculture.